# Structure-guided secretome analysis of gall-forming microbes offers insights into effector diversity and evolution

Soham Mukhopadhyay[1,2,3]*, Muhammad Asim Javed[1,2,3], Jiaxu Wu[1,2,3], Edel Perez-Lopez[1,2,3]*

[1]Département de Phytologie, Faculté des sciences de l'agriculture et de l'alimentation, Université Laval, Québec, Canada; [2]Centre de recherche et d'innovation sur les végétaux (CRIV), Université Laval, Québec, Canada; [3]L'Institute EDS, Université Laval, Québec, Canada

## eLife Assessment

This study presents an **important** discovery regarding the diversity and evolution of gall-forming microbial effectors. Supported by **convincing** computational structural predictions and analyses, the research provides insights into the unique mechanisms by which gall-forming microbes exert their pathogenicity in plants. This study also offers guidance that is of value for future studies on pathogen effector function and co-evolution with host plants.

*For correspondence:
soham.mukhopadhyay.1@ulaval.ca (SM);
edel.perez-lopez.1@ulaval.ca (EP-L)

**Competing interest:** The authors declare that no competing interests exist.

**Abstract** Phytopathogens secrete effector molecules to manipulate host immunity and metabolism. Recent advances in structural genomics have identified fungal effector families whose members adopt similar folds despite sequence divergence, highlighting their importance in virulence and immune evasion. To extend the scope of comparative structure-guided analysis to more evolutionarily distant phytopathogens with similar lifestyles, we used AlphaFold2 to predict the 3D structures of the secretome from selected plasmodiophorid, oomycete, and fungal gall-forming pathogens. Clustering protein folds based on structural homology revealed species-specific expansions and a low abundance of known orphan effector families. We identified novel sequence-unrelated but structurally similar (SUSS) effector clusters, rich in conserved motifs such as 'CCG' and 'RAYH'. We demonstrate that these motifs likely play a central role in maintaining the overall fold. We also identified a SUSS cluster adopting a nucleoside hydrolase-like fold conserved among various gall-forming microbes. Notably, ankyrin proteins (ANK) were significantly expanded in gall-forming plasmodiophorids, with most being highly expressed during clubroot disease, suggesting a role in pathogenicity. Subsequently, we screened ANK proteins against *Arabidopsis* immunity hubs using AlphaFold-Multimer and verified one of the positive results by Y2H and BiFC assays to show that the ankyrin effector PbANK1 targets host MPK3 and a zinc-binding dehydrogenase. These findings suggest a potential new mechanism in which ANK effectors target multiple host proteins involved in stress sensing, opening a novel avenue to study the role of ANK in host–pathogen interactions. Altogether, this study advances our understanding of secretome landscapes in gall-forming microbes and provides a valuable resource for broadening structural phylogenomic studies across diverse phytopathogens.

## Introduction

The evolutionary arms race between host plants and their microbial pathogens is a fascinating example of adaptation and counteradaptation. Central to this battlefield are the effectors — secreted proteins from pathogens that manipulate host cellular processes to facilitate infection and colonization (***Toruño***

**eLife digest** Microbes can cause a variety of plant diseases and pose a major threat to global food production. To infect plants, many of these microbes release small proteins called effectors. Once inside the plant cell, the effectors can disarm the plant's immune defences and also reprogram its growth. In some cases, they trigger abnormal swellings called galls, which can seriously reduce harvests, such as the clubroot disease of canola, caused by the protist *Plasmodiophora brassicae*.

Effectors produced by gall-forming microbes remain poorly understood because these pathogens are hard to grow in the laboratory, and many of their proteins have no known function. Mukhopadhyay et al. provide new insights into how gall-forming microbes infect plants and how their effectors have evolved using artificial intelligence tools, such as AlphaFold. These tools can predict the three-dimensional structures of proteins, allowing researchers to look beyond gene sequences and uncover hidden patterns in protein shapes.

To explore the molecular strategies used by different gall-forming organisms to infect plants, the researchers studied the three-dimensional structure and properties of thousands of secreted proteins from fungi, protists and oomycetes.

The results showed that each group showed unique expansions – that is, unusually large numbers – of particular effector families. For example, the clubroot pathogen had expanded families of ankyrin-repeat proteins, a class of proteins characterized by repeating structural motifs that mediate protein-protein interactions. They also discovered clusters of proteins that shared similar shapes despite having little or no genetic similarity, highlighting that protein structure can be more conserved than genetic sequence. Notably, one ankyrin-repeat protein was found to interact with central components of plant immunity, suggesting a direct role in disabling host defenses. Together, these findings provide the first structural map of effectors in gall-forming microbes.

By understanding how effectors work, researchers can identify plant genes that confer stronger resistance to pathogens such as the clubroot microbe. While more experiments are needed to confirm the roles of effectors in plants, this structural resource already offers a powerful tool for scientists. It could help predict which microbial proteins are most likely to manipulate plant health and guide the development of durable crop protection strategies.

*et al., 2016*). Effectors can also be recognized by plant receptor and trigger immunity and defense responses in the host (*Chen et al., 2022*). Due to their central role in pathogen-host interactions, effectors must continually evolve to evade detection by the host's immune system (*Martel et al., 2021*). This arms race drives rapid changes in effector sequences, often resulting in high mutation rates and diversification (*Liu et al., 2019*). Additionally, effectors should maintain structural integrity for functionality while altering surface residues to avoid immune recognition (*Derbyshire and Raffaele, 2023*). Recent studies have defined these groups of effectors as Sequence-Unrelated Structurally Similar (SUSS), which despite lacking sequence similarity, share significant structural homology (*Seong and Krasileva, 2021*; *Seong and Krasileva, 2023*). For example, the MAX effector family, which includes proteins from various fungal pathogens like *Magnaporthe oryzae* and *Pyrenophora tritici-repentis*, exhibits a conserved β-sandwich fold (*de Guillen et al., 2015*). Other examples include the RXLR-WY effector families in oomycetes (*Combier et al., 2022*; *Win et al., 2012*), the LARS effectors in *Cladosporium fulvum* and *Leptosphaeria maculans* (*Lazar et al., 2022*), the RALPH effectors in *Blumeria graminis* (*Cao et al., 2023*), and the FOLD effectors in *Fusarium oxysporum f. sp. lycopersici* (*Yu et al., 2024*) demonstrating structural conservation. FOLD effectors have also recently been found in the secretome of unrelated pathogenic and symbiotic fungi, pointing to the fold's relevance in plant colonization and expansion in different evolutive groups (*Teulet et al., 2023*). A recent study that classified orphan effector candidates (OECs) from Ascomycota into 62 main structural groups proposes that such structural conservation can be explained by changes in thermodynamic frustration at surface residues, which increase the robustness of the protein structure while altering potential interaction sites (*Derbyshire and Raffaele, 2023*).

Some of the discussed studies have been fueled by the emergence of machine-learning tools like AlphaFold (*Jumper et al., 2021*), which has revolutionized the field of protein modeling and enabled the computational prediction of pathogen effector structures. The utility of AlphaFold in

plant-microbe interactions has been further demonstrated by its ability to predict the structure of the highly conserved *AvrE*-family of effector proteins (*Nomura et al., 2023*). These proteins are crucial in the pathogenesis of various phytopathogenic bacteria but are challenging to study due to their large size, toxicity to plant and animal cells, and low sequence similarity to known proteins (*Xin et al., 2015*). The structure prediction revealed β-barrel structures similar to bacterial porins, allowing for modulating host cell functions by facilitating the movement of small molecules across plasma membrane. AlphaFold Multimer (*Evans, 2022*), an extension of AlphaFold designed to predict in-silico protein-protein interactions, has been recently used to identify 15 effector candidates capable of targeting the active sites of chitinases and proteases (*Homma et al., 2023*), expanding the applicability of these tools to advance the tailored functional characterization of effector proteins.

Recent advances in structure-guided secretome analysis have mostly focused on fungal pathogens due to their significant economic impact and the availability of robust genomic and transcriptomic resources. However, in recent years, there is a growing interest in protist pathogens, which are quickly becoming a threat to agriculture and environment (*Mukhopadhyay et al., 2024*). For example, the gall-forming *Plasmodiophora brassicae*, a protist belonging to the class Plasmodiophorid, can cause significant yield loss in canola fields (*Javed et al., 2023*; *Ochoa et al., 2023*). Moreover, these obligate biotrophic protists are impossible to culture in axenic media and therefore difficult to transform (*González-García and Pérez-López, 2021*). Although the effector repertory for some of these protists has been predicted, the majority of their secretome remains uncharacterized due to the absence of known protein domains (*Mukhopadhyay et al., 2024*). To gain more insights into the secretome composition and effector biology of these understudied pathogens, we conducted structural similarity-based clustering of the predicted effectors of selected plasmodiophorid, oomycete, and fungal gall-farming pathogens. Here we examined (*i*) if the primary secretome families present in each pathogen share common folds with known fungal effector families; (*ii*) if they share common folds that could be associated with their biotrophic lifestyle and gall-forming pathogenicity strategy; and (*iii*) if some of the known effectors from these pathogens are part of SUSS effector families. By comparing the secretome of gall-forming pathogens from distant lineages, we provide a comprehensive overview of the uniqueness and commonality of the secretome landscape and offer more insights into the protist effector families by bringing them into the structural genomics era.

## Results

### Secretome prediction and structural modeling of gall-forming pathogens

Based on genome availability, phylogenetic distance, and economic significance, six gall-forming pathogens were selected for secretome analysis: two plasmodiophorids, *Plasmodiophora brassicae* and *Spongospora subterranea*, oomycete *Albugo candida*, and three fungi from different lineages, *Taphrina deformans*, *Ustilago maydis*, and *Synchytrium endobioticum* (*Figure 1a*). To gain a better understanding of plasmodiophorids, for which structural data is scarce, we also included *Polymyxa betae*, a non-gall-forming plasmodiophorid vector of the beet necrotic yellow vein virus, causing Rhizomania disease (*Decroës, 2022*; *Figure 1a*). To identify the putative secretome of these plant pathogens, we first employed SignalP *Teufel et al., 2022* to predict sequences with N-terminal signal peptide. Sequences carrying a signal peptide were subjected to DeepTMHMM (*Hallgren et al., 2022*) search to remove sequences with transmembrane domains. Mature protein sequences greater than 1000 amino acid length were also removed (*Figure 1b*). The *Ustilago maydis* secretome and corresponding structures were obtained from a recent study (*Seong and Krasileva, 2023*) using similar filtering steps. This resulted in a total of 4197 proteins from seven gall-forming or related plant pathogens (*Supplementary file 1*). Next, the structures of 3575 proteins, excluding the *U. maydis* secretome which was already available, were modeled using AlphaFold2. InterProScan (*Jones et al., 2014*) was performed on all 4197 proteins against pfam, Gene3D, and SUPERFAMILY databases, identifying 41.59% of the proteins analyzed carrying a known protein domain (*Supplementary file 1*). Next, 2349 structures with pLDDT >65 were selected for further analysis (*Supplementary file 1*). Although pLDDT >70 is recommended as the threshold for structures with reliable confidence by AlphaFold developers, we selected a score of 65 or higher to include *AvrSen1* (pLDDT 67), the only known *S. endobioticum* avirulent gene (*van de Vossenberg et al., 2019a*; *Figure 1c*). This resulted in the

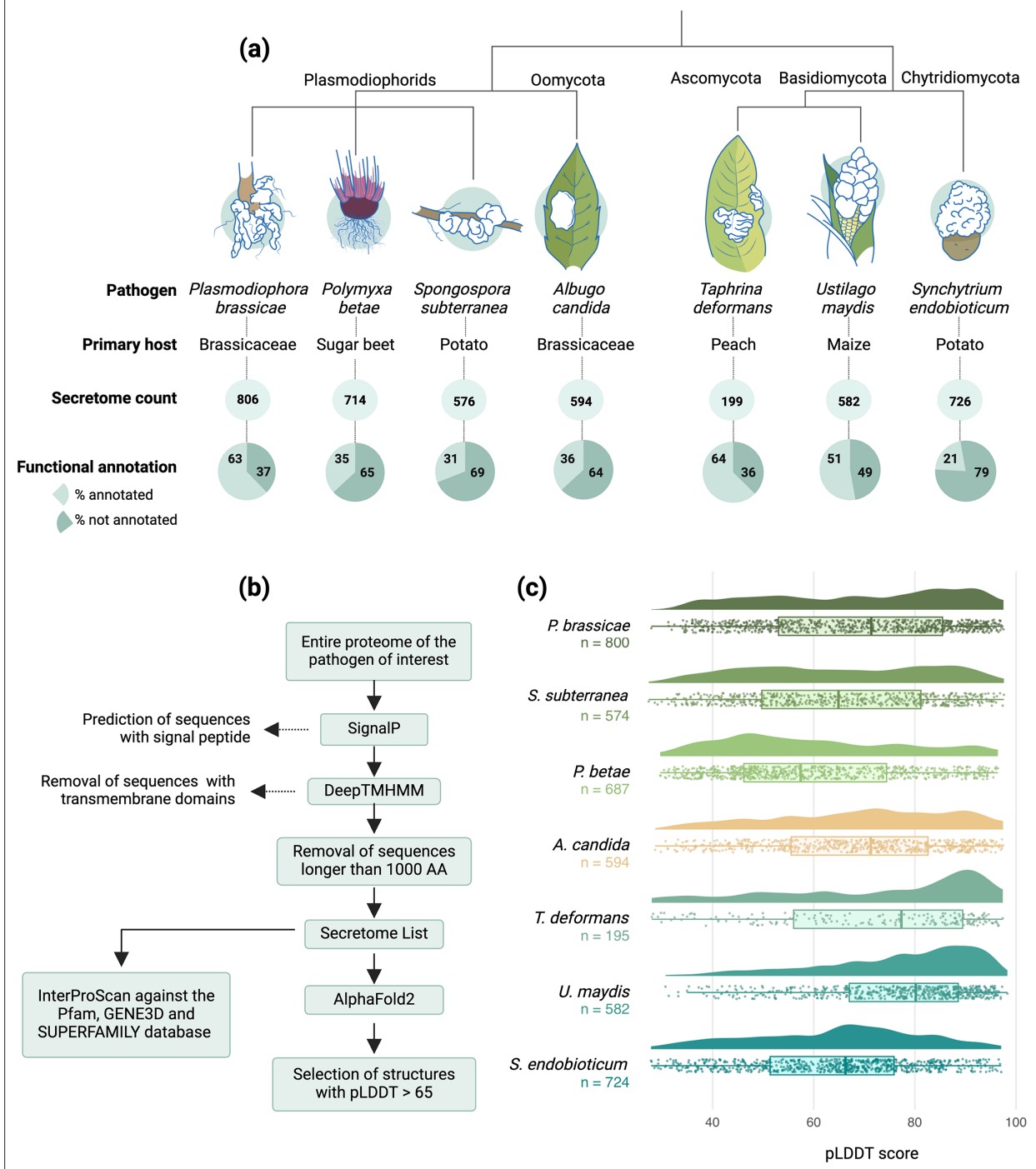

**Figure 1.** Description of the pathogens included in the study, workflow overview and statistics of structure prediction. (**A**) Cladogram of the pathogens used in the study. The schematics represent the disease symptoms on their respective hosts, with the white areas representing galls. The secretome count indicates the number of proteins per species predicted to be secreted, and the functional annotation shows the percentage of the secretome predicted to contain a known protein domain in the Pfam, SUPERFAMILY, or Gene3D databases. (**B**) Flowchart of the workflow used to predict the secretome and the corresponding 3D structures. (**C**) Raincloud plot showing the median and density distribution of pLDDT scores of the predicted structures in each pathogen.

inclusion of 298 candidate effectors which would have otherwise been excluded (*Supplementary file 1*). IUPred3 *Erdős et al., 2021* was also used to score the proteins as disordered or not based on whether 50% of the sequence position was predicted to be disordered (*Supplementary file 1*). Most of the proteins (97%) with pLDDT >70 were not disordered, while 26% of the secretome with

pLDDT <70 was disordered, thus showing the limitation of AlphaFold 2 in modeling such effectors (*Supplementary file 1*).

## Structure-based clustering reveals species-specific folds and low homology with known fungal effector families

Structural similarity among the predicted structures was assessed using TMAlign *Zhang and Skolnick, 2005* by scoring all structures against each other. To study the similarity with known effector families, we also included 19 crystal structures from the DELD, FOLD, LARS, MAX, KP6, RALPH, NTF2-like, ToxA, C2-like, and Zn-binding effector families, as previously utilized in a recent study (*Teulet et al., 2023*; *Supplementary file 2*). Additionally, we also included 62 structural families of orphan candidate effectors (OCE) recently identified in the Ascomycota lineage (*Derbyshire and Raffaele, 2023*; *Supplementary file 2*). Comparisons with TMScore greater than 0.5 were considered positive for structural similarity. Markov clustering (*Enright et al., 2002*; *Van Dongen, 2008*) with an inflation value of two was applied to cluster the secretome based on structural similarity score. This resulted in 254 structural clusters with at least two members (*Figure 2*, *Supplementary file 3*).

Ankyrin repeat-containing proteins were the largest cluster for plasmodiophorid *P. brassicae* (n=42) and *S. subterranea* (n=39) (*Figure 2*). The largest cluster detected on *A. candida* secretome is formed by 'CCG' (*Furzer et al., 2022*) class of effectors (n=48), while *U. maydis* largest cluster was composed of the Tin2-like proteins (*Tanaka et al., 2019*) (n=31). For *S. synchytrium*, the largest cluster (n=64) contains AvrSen1 virulence factor (*Figure 2a*). *P. betae* and *T. deformans* do not carry large (n>30) effector clusters which could be due to *P. betae*'s vector-like nature and *T. deformans*'s reduced genome size compared to other fungi (*Cissé et al., 2013*). *T. deformans*'s largest cluster is primarily composed of various glycoside hydrolases, while *P. betae*'s largest cluster consists of orphan helical proteins (*Supplementary file 3*). A 20-member kinase family (cluster 13) was noted in *P. brassicae*, which was absent in *A. candida* and three other fungi. Chitin deacetylases, which have been reported to convert chitin to less immunogenic chitosan (*Gao et al., 2019*), were expanded (n=11) in *P. brassicae* (*Supplementary file 3*). Chitin deacetylases were found in five out of seven pathogens tested, except in oomycete *A. candida* and ascomycete *T. deformans*, which despite being a fungus contains very little chitin in the cell wall (*Petit and Schneider, 1983*; *Supplementary file 3*). In *S. subterranea* and *P. brassicae*, a unique TauD/TfdA protein cluster was also identified, typically found in bacteria for taurine utilization as a sulfur source (*Eichhorn et al., 1997*). *P. betae* carries a six-member G-domain protein cluster, with similarity to its host *Beta vulgari*'s GTPases (*Figure 2—figure supplement 1*, *Supplementary file 3*). Among the well-characterized fungal effector families, only three KP6 fold *Thynne et al., 2024* and one RALPH-like fold were found in *U. maydis* (*Figure 2—figure supplement 1*, *Supplementary file 3*). *T. deformans* carries two ToxA homologs (*Figure 2—figure supplement 1*, *Supplementary file 3*).

## Effector folds conserved across kingdoms

Six protein folds, Hydrolases (clusters 2, 8, 9), Carboxypeptidases (cluster 12), Aspartyl proteases (cluster 3), Lectins (cluster 38), SCP domain (cluster 24), and an orphan group (cluster 5), contained proteins from all the pathogens investigated in this study, indicating deep evolutionary conservation of these folds across kingdoms (*Supplementary file 3*). In fact, 20 out of 62 Ascomycete orphan effector groups had at least one structural homolog in plasmodiophorids and oomycetes tested in this study, although a complete evolutionary connection would require the comparison of a much larger number of pathogens (*Supplementary file 3*). We did not find any specific fold (n>1 per species) conserved only among the gall-forming pathogens studied, indicating that this virulence strategy can be achieved by different mechanisms without necessarily converging onto common effector folds.

## Structural search identifies a nucleoside hydrolase-like fold conserved in some gall-forming pathogens

Effectors are notorious for carrying unknown domains, making it difficult to predict the putative function of promising candidates (*Lovelace et al., 2023*). We searched for uncharacterized *P. brassicae* candidate effectors within the same cluster that were also overexpressed during infection. Two candidate effectors, PBTT_09143 and PBTT_07479, which are the first and fifth most expressed proteins at 16 days post inoculation (dpi) during clubroot disease, belong to the cluster 21 grouping with

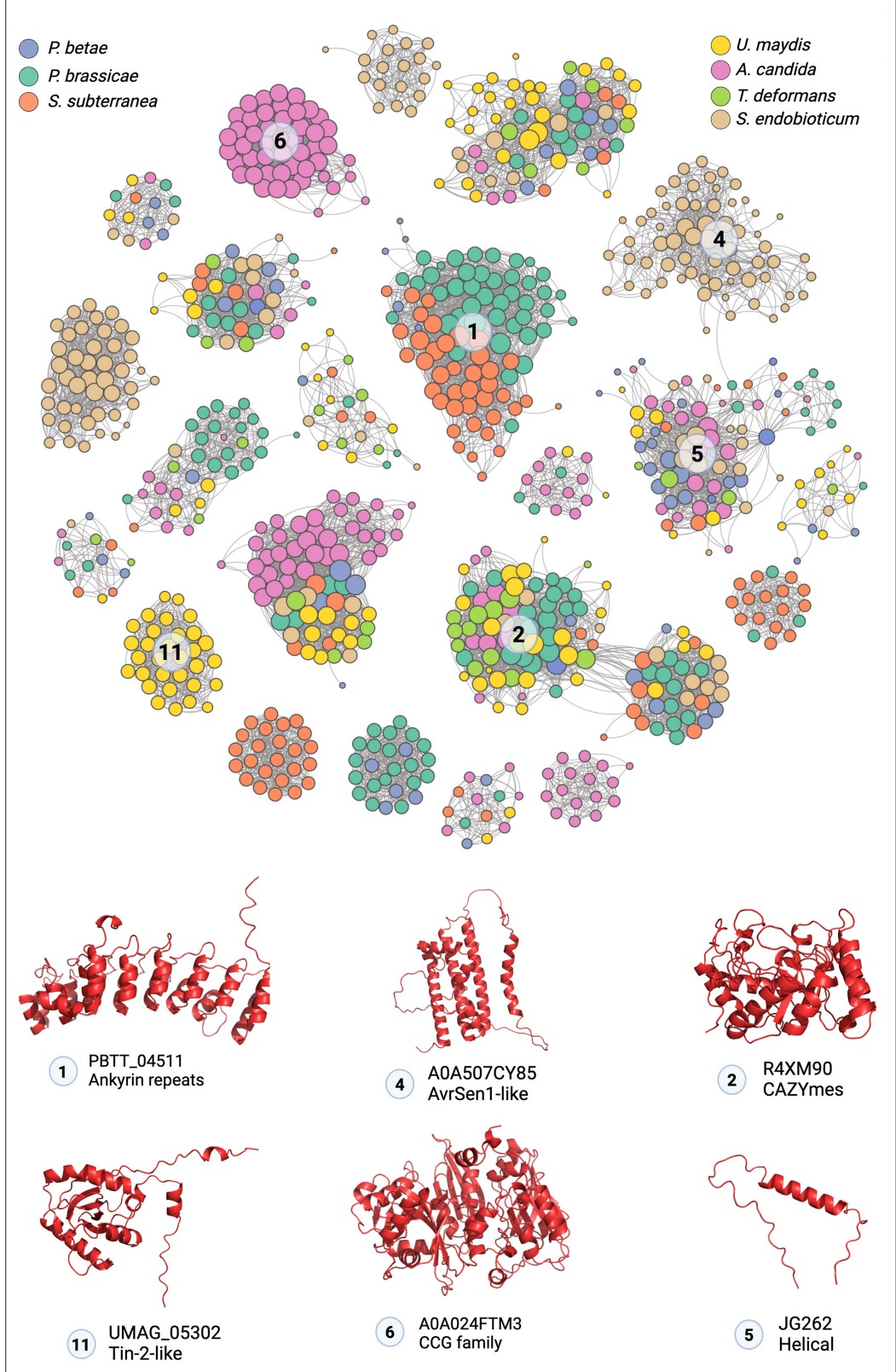

**Figure 2.** Visualization of dominant protein folds present in each pathogen. (Top) Network plot of structurally similar secretome clusters with at least 15 members. Not all 255 clusters are shown to reduce complexity. Each node represents a single protein, and an edge between two nodes represents structural similarity (TMScore >0.5).

*Figure 2 continued on next page*

*Figure 2 continued*

(Bottom) Representative structure of the dominant fold in each pathogen. Since Ankyrin repeats are common in both *P. brassicae* and *S. subterranea*, they are represented only once.

The online version of this article includes the following figure supplement(s) for figure 2:

**Figure supplement 1.** The structures of *Beta vulgari*'s GTPases, KP6 fold, RALPH-like fold, and ToxA fold share structural similarity with candidate effectors found in *P. betae*, *U. maydis*, and *T. deformans*, respectively.

PBTT_0412, none of which carries a known domain (*Figure 3a–c*). Interestingly, the cluster also includes ten members from *A. candida* secretome, with eight carrying a predicted nucleoside hydrolase domain (*Supplementary file 4*). We subjected the three *P. brassicae* candidates to a FoldSeek-mediated (*van Kempen et al., 2024*) structural search against the PDB100 and AFD-proteome databases. Nucleoside hydrolases always emerged as the top hit (E-value <10^−5, Prob ~1, TMscore ≥ 0.5; *Supplementary file 4*). Next, searching the AFDB cluster web tool (*Barrio-Hernandez et al., 2023*), which allows for the identification of structural homologs across the known protein space, AFDB clusters A0A0G4IP88 and A0A024FV66 emerged as the top hits. The members of these clusters are often gall-forming, carry predicted signal peptide, and belong to various biotrophs *like Melanopsichium pennsylvanicum, Ustilago maydis, Albugo candida, Sporisorium scitamineum, Testicularia cyperi, Colletotrichum orbiculare*, among others (*Figure 3b and d*). Some of the members also do not carry identifiable domains (*Supplementary file 4*) and show limited sequence similarity among themselves (*Figure 3b, Supplementary file 4*). Interestingly, the cluster also includes bacterial Type

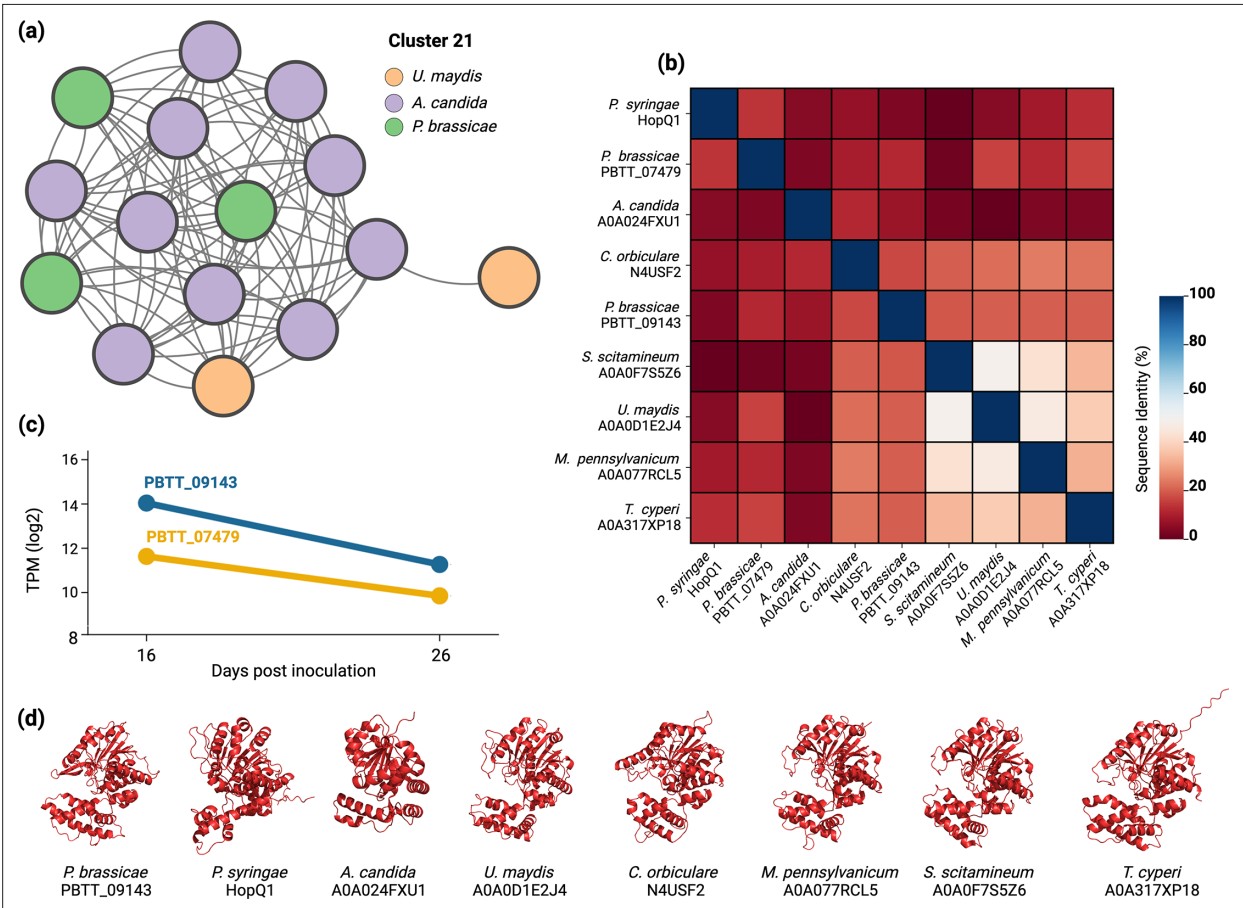

**Figure 3.** Sequence and structural similarity among HopQ1 homologs in *U. maydis*, *A. candida*, and *P. brassicae*. (A) Network plot showing the structural similarity between the members of cluster 21. Edges denote structural similarity (TMScore >0.5). (B) Pairwise sequence identity between selected HopQ1 structural homologs from plasmodiophorids, oomycetes, and fungi, illustrating sequence dissimilarity between some proteins despite structural homology. (C) Gene expression values (log2 TPM) of two highly induced *P. brassicae* genes at 16 and 26 dpi. (D) 3D structure of the mature protein sequences, assuming a HopQ1-like fold.

III effector HopQ1 from *Pseudomonas syringae* (*Supplementary file 4*). Unlike *P. brassicae* effectors, HopQ1 is predicted to carry a nucleoside hydrolase domain, although the domain has been reported to be unable to bind standard nucleosides (*Li et al., 2013*). HopQ1 has been reported to be associated with 14-3-3 plant proteins to promote virulence (*Li et al., 2013*). Thus, it's possible that the nucleoside hydrolase-like fold present in various gall-forming fungal, protist, and oomycete pathogens might have also neo-functionalized and be involved in new molecular strategies during the infection.

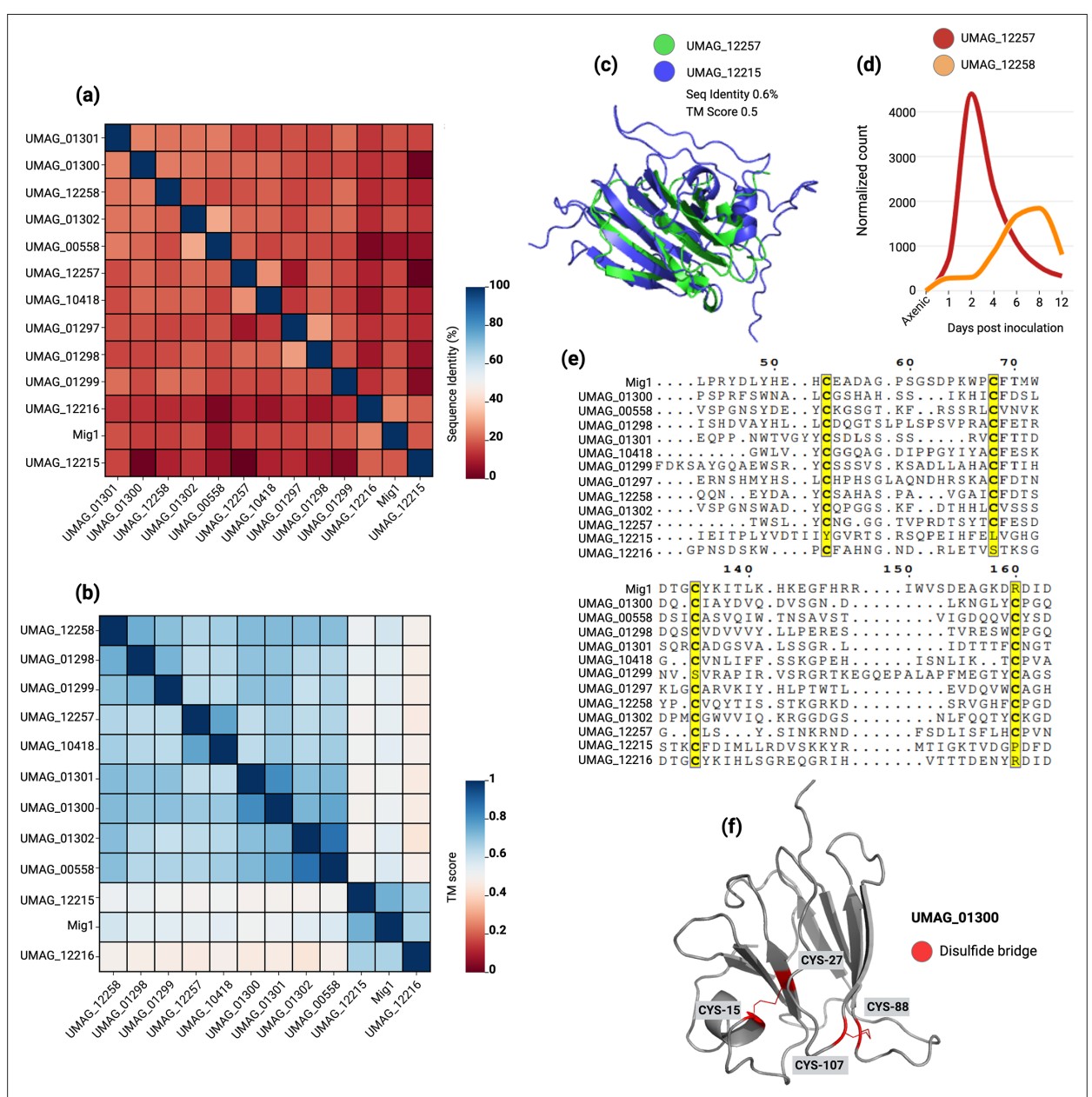

**Figure 4.** Sequence and structural similarity among Mig1 homologs in *Ustilago maydis*. (**A**) Similarity matrix showing the pairwise sequence identity (%) between Mig1 cluster members. (**B**) Similarity matrix showing the pairwise structural homology scores (TMScore) between Mig1 cluster members. (**C**) Superimposition of two Mig1 homologues, illustrating structural similarity despite extreme sequence divergence. (**D**) Differential gene expression patterns of two Mig1 tandem duplicates. (**E**) Multiple alignment of protein sequences, highlighting the conservation of cysteine residues (marked in yellow). (**F**) Visualization of the conserved cysteine residues forming disulfide bridges.

## A fungal effector family shows structural homology despite extreme sequence divergence

The Mig1 protein in *Ustilago maydis* is a maize-induced effector that plays a crucial role in the biotrophic interaction between the fungus and its host (*Basse et al., 2000*). It is specifically induced during the biotrophic phase, contributing to the fungus's pathogenicity. Cluster 30, specifically found in *Ustilago maydis*, contains this effector (*Supplementary file 5*). Upon examining the sequence and structural similarities between the members of the cluster, we discovered instances of structural homology (TMScore >0.5) despite pairwise sequence identity being as low as 0.6% and no higher than 30% (*Figure 4a–d*, *Supplementary file 5*). When aligning the protein sequences of 13 members, four cysteine residues were found to be strongly conserved (*Figure 4e*). These four residues form two disulfide bridges (*Figure 4f*), likely playing a crucial role in maintaining the overall fold despite significant sequence dissimilarity. All 13 members of the cluster were expressed upon infection (*Supplementary file 6*). A variable expression was observed for the *Mig1*-like genes located in the same genomic region, including members with completely opposite patterns of induction (*Supplementary file 6*), suggesting a possible regulatory role or the acquisition of new functions.

## Identification of new SUSS effector families enriched in known motifs

The fungi *S. endobioticum* and *A. candida* encode large effector clusters which included previously identified avirulent factors AvrSen1 and CCG28/31/33/70, respectively (*Supplementary file 7*; *Redkar et al., 2023*). It was previously shown that the N-terminal region of the CCG28,33 and 70 share structural homology (*Redkar et al., 2023*). To examine if these clusters represent SUSS families, we carried out the sequence-based clustering of the 4197 proteins from the seven pathogens investigated here. We performed a BlastP search in all-vs-all mode and kept only those results with E-value lower than 10^−04 and bidirectional coverage of 50%. Markov clustering revealed 642 sequence-based clusters with at least two members (*Supplementary file 7*). We searched for sequence clusters having members from the same structural clusters. This revealed the presence of 12 sequence-related clusters associated with the AvrSen1 structural cluster and 11 sequence-related clusters associated with the CCG structural cluster (*Figure 5a*, *Supplementary file 7*). It was previously reported that CCG-containing effectors share limited similarity around the CCG motif and can be grouped in several clades based on sequence similarity (*Furzer et al., 2022*). Thus, AvrSen1 and CCG represent novel SUSS effector families whose similarities can't be delineated by sequence search alone. Integration of sequence and structure data increased the member count of AvrSen1 and CCG clusters to 124 and 50 from previous 64 and 48, respectively (*Supplementary file 8*).

It has been reported that *S. endobioticum* secretome is enriched in RAYH (*van de Vossenberg et al., 2019b*). While it is understood that the CCG class of effectors derived its name due to the presence of the CCG motif, it was not immediately clear if Avrsen1 cluster was also the source of the conserved RAYH motif. To verify that, we subjected the mature protein sequences of the two pathogen secretome to motif search using MEME (*Bailey and Gribskov, 1998*). Here we identified a 16 amino acid long RAYH motif present in 118 *S. endobioticum* proteins and 15 amino acid long 'CCG' motif present in 74 *A. candida* proteins (E<0.1, combined match p<0.001; *Figure 5b*, *Figure 5—figure supplement 1*, *Figure 5—figure supplement 2*, *Supplementary file 8*). Of the 118 sequences proteins carrying the RAYH motif, 79 were members of the AvrSen1 cluster in *S. endobioticum* (*Supplementary file 8*). Thus, the expansion of SUSS effectors in *A. candida* and *S. synchytrium* has resulted in the enrichment of common motifs, something that has recently been observed for the Y/F/WxC motif in *Blumeria graminis* RNA-like effector cluster (*Seong and Krasileva, 2023*).

Selection pressure analysis on *A. candida* CCG members shows that both cysteine residues in the 'CCG' motifs and two additional cysteine residues within 50 amino acids of the motif are often under purifying selection (*Figure 5—figure supplement 3*, *Figure 5—figure supplement 4*). Visualizing these four cysteines on the predicted structure shows that they form disulfide bridges and probably play a crucial role in overall maintenance of the fold (*Figure 5c*). The CCG motif seems to be a crucial part of a module consisting of two parallel alpha-helices joined to a beta-sheet, and CCG effectors are often composed of several of these modules (*Figure 5c*). The RAYH motif was also found to be part of the core structure of most AvrSen1-like effectors, forming a long alpha-helix (*Figure 5d*). Apart from the RAYH motif region, several surrounding hydrophobic residues were also strongly conserved, likely playing a role in maintaining the structure (*Figure 5—figure supplement 5*). Examining the

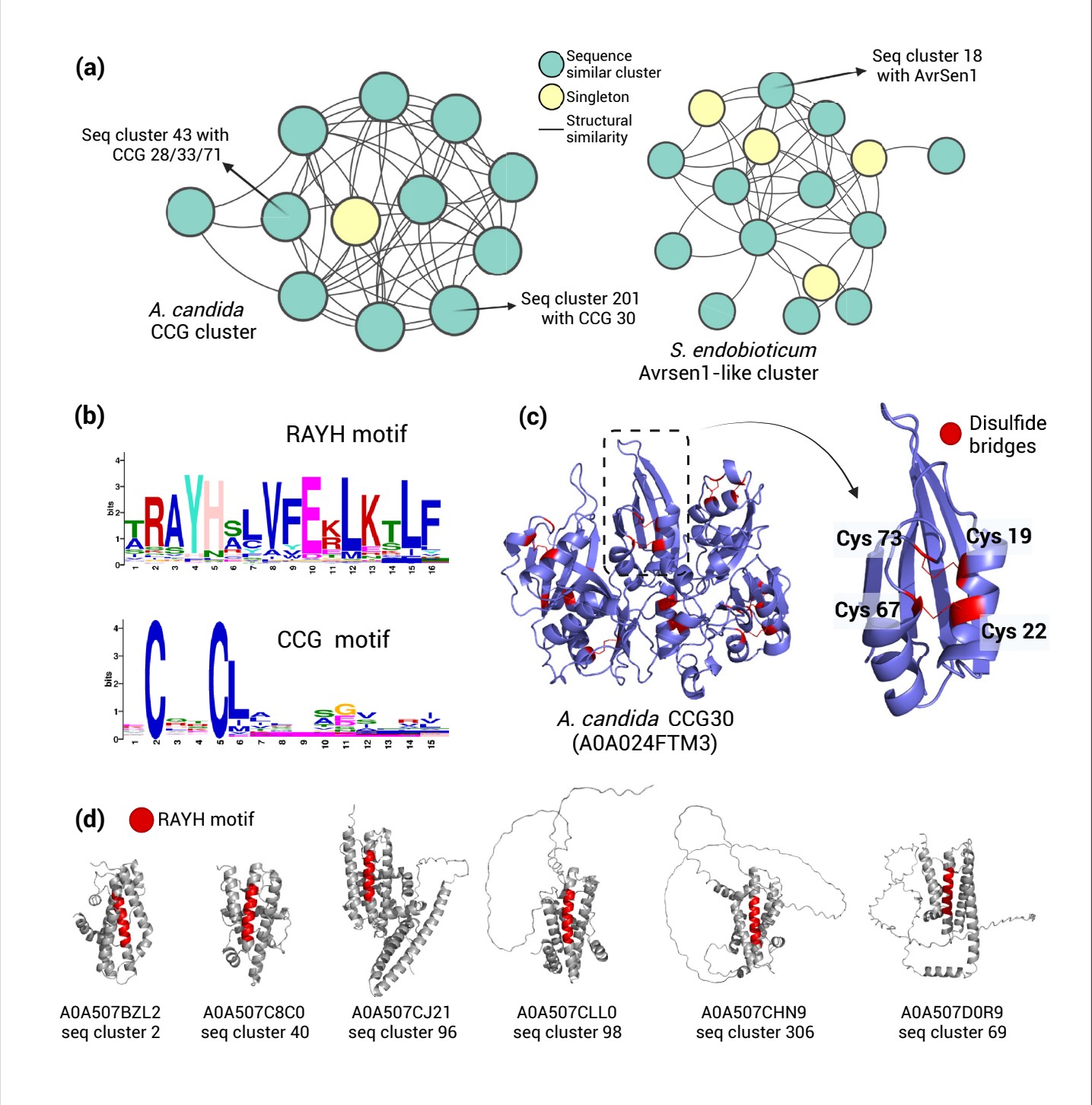

**Figure 5.** SUSS effector families are enriched in common motifs. (**A**) Network plots demonstrating that the two primary effector families in *A. candida* and *S. endobioticum* can only be grouped together when structural data is incorporated into the sequence-based clustering. The plots also indicate which sequence-based clusters contain the known effectors from these groups. (**B**) 'RAYH' and 'CCG' motif patterns identified by MEME scan. (**C**) Disulfide bridges in the 'CCG' motif, likely playing a pivotal role in structural maintenance, are highlighted in the virulence factor CCG30. A zoomed-in view of the 'CCG module' shows the four conserved cysteine residues forming disulfide bridge. (**D**) The 'RAYH' motif, occupying the central position in the core alpha-helix bundle, is highlighted in six sequence-based subclusters within the AvrSen1-like cluster in *S. endobioticum*.

The online version of this article includes the following figure supplement(s) for figure 5:

**Figure supplement 1.** Location of CCG motifs identified by MEME scan.

*Figure 5 continued on next page*

*Figure 5 continued*

**Figure supplement 2.** Location of RAYH motifs identified by MEME scan.

**Figure supplement 3.** Analysis evidencing purifying selection in the CCG effector family.

**Figure supplement 4.** Multiple alignment of representative members of CCG sequence-based clusters.

**Figure supplement 5.** Multiple alignment of representative members of AvrSen1-like sequence-based clusters.

sequence-related subclusters of the AvrSen1-like family, we found that the effectors are evolving by keeping the core alpha-helix bundle fixed while diversifying the peripheral stretches (*Figure 5d*).

## Ankyrin repeat-containing proteins are a common feature of gall-forming plasmodiophorids

The largest structural cluster in *P. brassicae* and *S. subterranea* consists of ankyrin repeat-containing proteins (hereafter ANK proteins; *Figure 6a*). The presence of only five members in the non-gall-forming plasmodiophorid *P. betae*, in contrast to over 40 members in the gall-forming *P. brassicae* and *S. subterranea*, highlights the importance of this domain in their pathogenicity strategies (*Supplementary file 9*). Interestingly, InterProScan identified 32 additional *P. brassicae* proteins with ANK domains, while only two *S. subterranea* ANK proteins could be identified outside of the cluster (*Supplementary file 9*). We found that the *P. brassicae* secretome is richer in repeat proteins compared to *S. subterranean* ANK proteins repertory (*Figure 6a*), with 19 additional leucine-rich repeats (LRRs), ten tetratrico-peptide repeats, and three MORN repeats (*Supplementary file 9*). Although 40 out of 74 ankyrins are overexpressed (TPM >10) at 16- and 26 dpi in *Arabidopsis thaliana*, only five LRRs are induced (*Supplementary file 9*). Notably, 17 ankyrin-repeat proteins also carry the SKP1/BTB/POZ domain, which is often involved in ubiquitination (*Geyer et al., 2003*; *Figure 6b*, *Supplementary file 9*).

A MEME motif scan with the members of the ankyrin cluster identified a 33 amino acid long motif in *P. brassicae* and a 32 amino acid length motif in *S. spongospora* (*Figure 6c*, *Supplementary file 9*). Aligning the MEME profile of the two identified ankyrin motifs shows a strong conservation of two leucine and one alanine residues (*Figure 6d*) and upon visualization, those residues form the hydrophobic pocket between the two alpha helices of the ankyrin repeat, stabilizing the structure (*Figure 6d*). The rest of the non-conserved residues were found to be highly polymorphic (*Figure 6d*).

## Ankyrin repeat-containing proteins interact with multiple host targets

Finally, to have an idea of the possible role of ANK proteins in plasmodiophorid virulence, we selected 70 ankyrin domain-containing proteins from *P. brassicae* and screened them against 20 key immune-related genes in *Arabidopsis* using AlphaFold-Multimer (*Supplementary file 10*). Protein-protein interactions were considered significant if the inter-chain Predicted Aligned Error (PAE) value was below 10, and the iPTM +pTM score was 70 or higher. Among the identified interactions, MPK3, MAPK4, MAPK6, SnRK1, NPR1, XCP1, CNGC4, and BAK1 were targeted by a total of ten ankyrin domain proteins (*Figure 6e*, *Supplementary file 11*). This dataset should serve as a valuable starting point for further understanding the role of ANKs in the virulence mechanisms of plasmodiophorids.

To experimentally validate the AlphaFold-Multimer predictions, we selected the PBTT_00818 (hereafter PbANK1)–MPK3 pair for targeted yeast two-hybrid (Y2H) assays. This pair was prioritized due to its high combined iPTM +pTM confidence score (0.82). Surprisingly, the one-on-one Y2H assay did not detect an interaction between PbANK1 and MPK3 (*Figure 7a*). To broaden our search, we next employed a large-scale Y2H screen using PbANK1 as bait against a randomly primed *Arabidopsis* seedling cDNA library, screening approximately 60 million clones. From this screen, 98 His[+] colonies were recovered on selective medium lacking tryptophan, leucine, and histidine and supplemented with 50 mM 3-aminotriazole to suppress bait autoactivation. Among the recovered clones, several candidate interactors of PbANK1 were identified, with the GroES-like zinc-binding alcohol dehydrogenase (AT3G56460) emerging as the top candidate (*Supplementary file 12*). The PBTT_00818–GroES-like interaction was subsequently confirmed via one-on-one Y2H assay (*Figure 7b*). To further investigate these interactions in planta, we performed bimolecular fluorescence complementation (BiFC) assays in *Nicotiana benthamiana* leaves. These assays confirmed that PbANK1 interacts with both MPK3 (*Figure 7c*, *Figure 7—figure supplement 1*) and GroES-like (*Figure 7d*, *Figure 7—figure supplement 2*), with both interactions occurring in the nucleus. Together, these findings illustrate

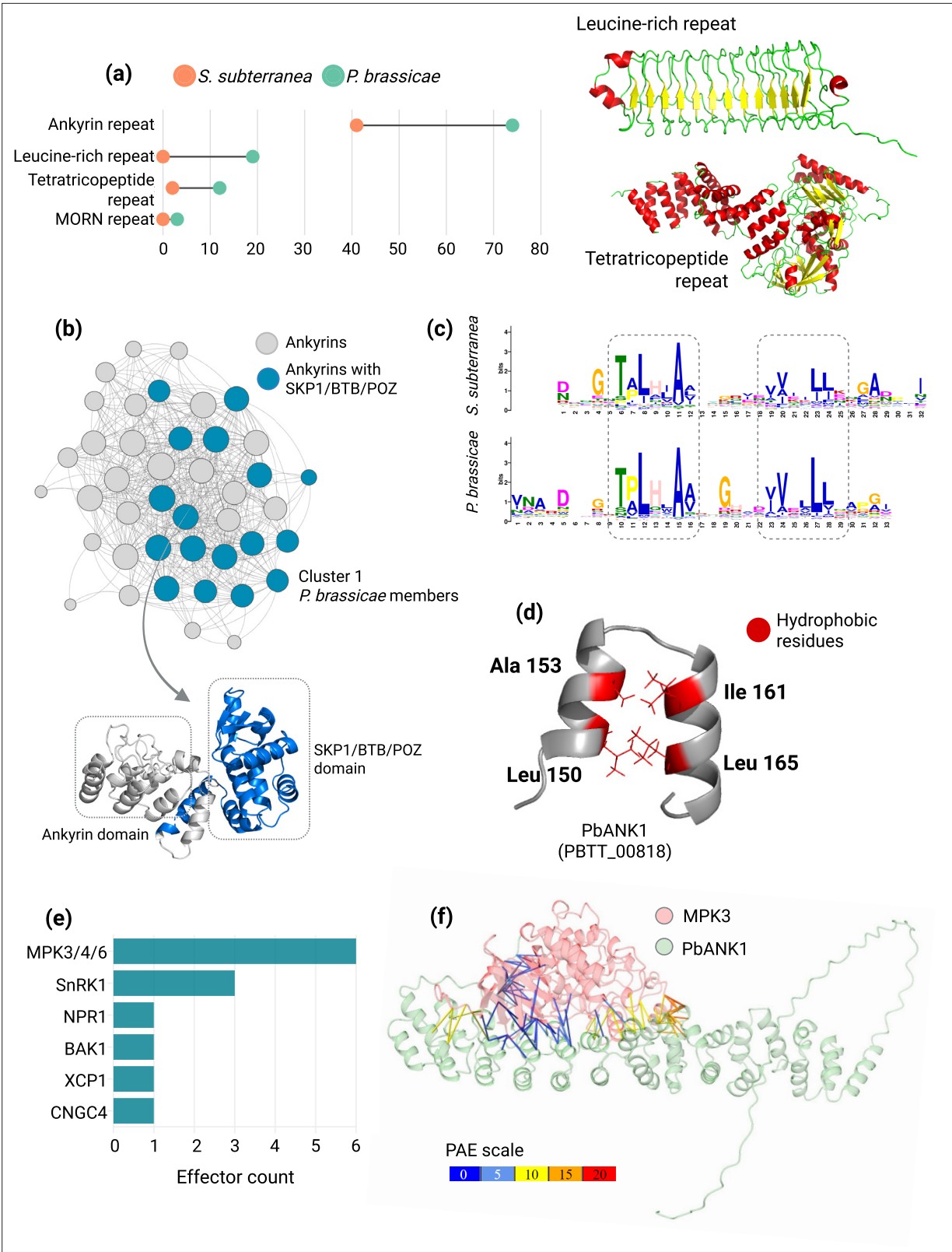

**Figure 6.** Diversity, structural features, and host immune targets of ankyrin repeats in Plasmodiophorids. (**A**) Frequency of repeat-containing proteins in *P. brassicae* and *S. subterranea*. (**B**) Network plot showing structural homology within *P. brassicae* Ankyrin repeats, also highlighting the Ankyrin repeats with SKP1/BTB/POZ superfamily domains. (**C**) Alignment of Ankyrin motifs from *P. brassicae* and *S. subterranea*. (**D**) Visualization of conserved hydrophobic residues in a single Ankyrin repeat module. (**E**) Number of Ankyrin repeat proteins predicted to target *Arabidopsis* immune proteins. (**F**)

*Figure 6 continued on next page*

*Figure 6 continued*

AlphaFold Multimer predicted complex of MPK3 and PbANK1 (PBTT_00818), highlighting the predicted aligned errors of surface contacts under 4 Ångströms.

how in silico protein–protein interaction predictions can serve as a powerful tool to generate testable hypotheses about effector functions in host-pathogen systems.

## Discussion

This study identified the primary protein folds in gall-forming pathogens' secretome supporting the idea that pathogen secretome is often dominated by expansion of specific folds which have been adopted and diversified over the course of evolution. Characterizing these primary effector folds in understudied plasmodiophorids like *P. brassicae* and *S. subterranea* would offer valuable insights into their virulence strategies, which remain largely enigmatic (*Mukhopadhyay et al., 2024*; *Pérez-López et al., 2018*). Here, we found that the ankyrin proteins, which are significantly expanded in gall-forming plasmodiophorids but less so in the related species *P. betae*, may be key to their ability to manipulate host immune responses and promote gall formation (*Figures 2 and 6*). This finding aligns with previous research indicating the importance of repeat-containing proteins in the virulence of plant pathogens (*Mesarich et al., 2015*). Examples of that are *Phytophthora* spp. effectors containing tandem repeats of the '(L)WY' motif, whose modularity and elaborate mimicry of a host phosphatase helps to promote infection (*Li et al., 2023*). ANK motifs are well-known for mediating protein-protein interactions (*Li et al., 2006*), and have been identified as type IV effectors in the intracellular human pathogens *Legionella pneumophila* and *Coxiella burnetii* (*Pan et al., 2008*). Thus, based on the predicted structure and the validated interactions, we hypothesize that the highly polymorphic surface residues of plasmodiophorid ANK motifs, along with their variable frequency of occurrence across effector proteins, result in diverse binding interfaces for distinct host targets. However, further studies are needed to elucidate the mechanistic basis of how ANK proteins can engage multiple targets, including those reported here like MPK3 involved in immunity *Bradley et al., 2022* and GroES-like protein implicated in peroxisomal functions (*Xu et al., 2016*).

This study also identified conserved protein folds across multiple kingdoms, particularly the hydrolase, carboxypeptidase, and aspartyl protease folds (*Supplementary file 3*). These folds appear to play a fundamental role in the virulence strategies of these pathogens, likely due to their ability to perform essential biochemical functions that facilitate infection (*Reumann et al., 2007*). The conservation of these folds across diverse species highlights their evolutionary significance and suggests that they may represent mechanisms of host manipulation that have been retained through speciation. We also identified a nucleoside hydrolase-like fold in evolutionary distant gall-forming pathogens which has homology to bacterial effector HopQ1 (*Figure 3*). Nucleoside hydrolases are involved in the purine salvage pathway in various pathogens (*Hofer, 2023*), but similar to HopQ1's mode of action, these gall-forming biotrophs might also be targeting 14-3-3 proteins, which are implicated in hormonal signaling (*Camoni et al., 2018*). Although HopQ1 is widely conserved within the *Pseudomonas* species complex, our FoldSeek-mediated search identified hits in *Pseudomonas savastanoi* isolates, some of which are known to form galls on woody plants (*Harmon et al., 2018*; *Ramos et al., 2012*). It remains to be seen if some of the HopQ1 homologs have been specifically adapted in these bacteria to support a particular lifestyle.

Here we also provided further evidence for the divergent evolution model of effector evolution, which describes how members of the same effector family can exhibit extreme sequence dissimilarity over a long period while retaining the core fold intact (*Seong and Krasileva, 2023*). We show that this evolution mechanism occurs in both fungi and oomycetes and often involves the conservation of cysteine or hydrophobic residues to maintain the original fold (*Figure 4*, *Figure 5*). Given that sequence dissimilarity between homologs can be extreme, as exemplified by the Mig1 family in *U. maydis*, it would become common practice among researchers to incorporate structural knowledge into sequence searches to accurately gauge the diversity of the effector families (*Figure 4*).

Overall, our study underscores the power of structural genomics and machine learning tools like AlphaFold2 in uncovering the complexities of pathogen effector repertoires. The findings presented here open new avenues for research into the evolution of virulence strategies in phytopathogens and

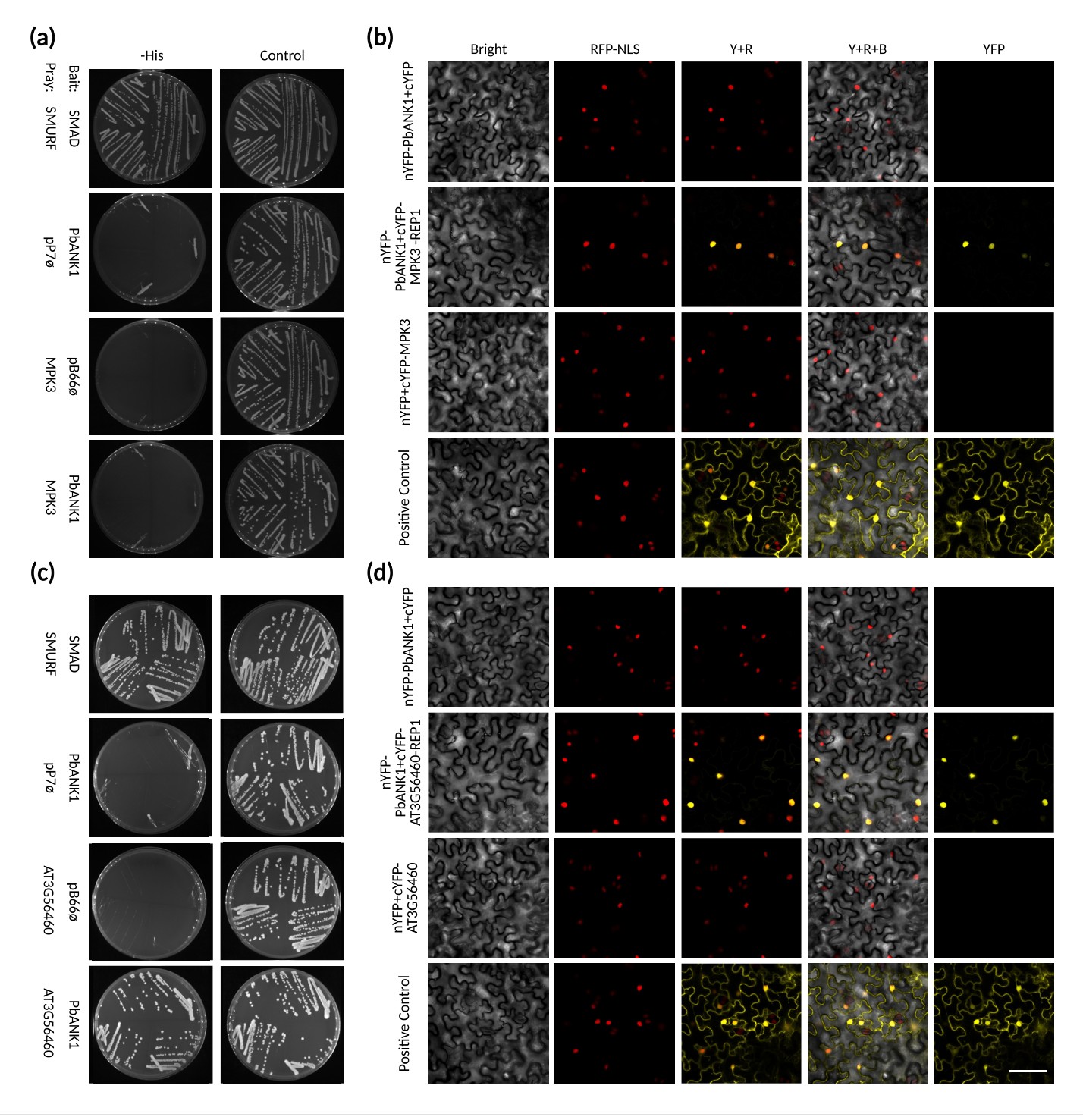

**Figure 7.** Validation of PbANK1-MPK3 and PbANK1-GroES-like interactions through Yeast two-hybrid (Y2H) and bimolecular fluorescence complementation (BiFC). (**A**) 1-by-1 Y2H assay results evaluating the interaction PbANK1-MPK3 (AT3G45640) predicted through AlphaFold-Multimer. N=3. (**B**) 1-by-1 Y2H assay results evaluating the interaction PbANK1-GroES-like (AT3G56460) predicted through a Y2H screening of an *Arabidopsis* seedling library. N=3. (**C**) BiFC assay results evaluating the interaction PbANK1-MPK3. N=3, presented in *Figure 7—figure supplement 1*. Bar = 50 μm. (**D**) BiFC assay results evaluating the interaction PbANK1- GroES-like. N=3, presented in *Figure 7—figure supplement 2*. Bar = 50 μm.

The online version of this article includes the following figure supplement(s) for figure 7:

**Figure supplement 1.** Extended BiFC assay results evaluating the interaction PbANK1-MPK3 presented in *Figure 7c* including the three replicates, empty vectors, and positive controls.

*Figure 7 continued on next page*

*Figure 7 continued*

**Figure supplement 2.** Extended BiFC assay results evaluating the interaction PbANK1-GroES-like presented in *Figure 7d* including the three replicates, empty vectors, and positive controls.

highlight the potential for these insights to inform the development of novel approaches to plant disease management. As we continue to expand our understanding of effector biology, particularly in under-studied pathogens, it will be crucial to integrate these structural insights with functional studies to fully elucidate the roles of these proteins in host-pathogen interactions.

# Materials and methods

## Secretome prediction

The proteome of *P. brassicae* was derived from our recent study generating the first complete genome of the clubroot pathogen (*Javed et al., 2024*). We are thankful to Prof. Anne Legrève for providing the updated annotation of the *P. betae* genome and proteome previously published by them (*Decroës et al., 2019*; *Decroës et al., 2022*). The rest of the proteome for *S. subterranea* (*Ciaghi et al., 2018*), *A. candida* (*McMullan et al., 2015*), *S. endobioticum* (*van de Vossenberg et al., 2019b*), *U. maydis* (*Kämper et al., 2006*), and *T. deformans* (*Cissé et al., 2013*) were downloaded from Uniprot database. SignalP 6 was utilized to identify sequences with predicted signal peptides, which were subsequently removed. DeepTMHMM was run through the pybiolib package. InterProScan 5.61–93.0 was used to confirm the presence of known domains using the Pfam, Gene3D, and SUPERFAMILY databases.

## Structure prediction

A total of 3615 mature protein sequences were selected to be modeled using AlphaFold 2, but 40 predictions repeatedly failed at the MSA construction step, resulting in 3575 structures. To expedite the process, ParaFold 2.0 (*Zhong et al., 2022*) was used, which employs AlphaFold 2.3.1 internally but distributes the CPU and GPU tasks to facilitate parallelization. 'Valeria' compute cluster (https://valeria.science/accueil) of Université Laval was used for the structure prediction. The full database was used to construct the MSA (multiple sequence alignment), and models were predicted in 'monomer' mode, resulting in five PDB structures sorted by pLDDT scores. The Rank_0 PDB was used for subsequent studies. Finally, 2000 models with pLDDT scores over 65 were selected for downstream analysis.

## Similarity search, clustering, and network plots

TM-Align was used to perform an all-versus-all structural comparison of 2000 models, and those comparisons with a normalized TM-score above 0.5 were considered significant. All-versus-all sequence comparison was performed using BlastP (*Camacho et al., 2009*) with an E-value <10^–4 and bidirectional coverage of at least 50%. Structure and sequence similarity data, represented by three columns with the first two as target and source IDs, and the third one being TM-score/E-value, were clustered using the Markov clustering with an inflation value of two. For sequence clustering, E-values were loaded following the recommendation of the MCL workflow [mcxload `-abc seq.abc --stream-mirror --stream-neg-log10 -stream-tf 'ceil(200)'`]. Custom Python scripts were written to find the sequence-related subclusters belonging to the same structural cluster and to count the occurrence of cluster members (https://github.com/Edelab/AlphaFold_effector_paper, copy archived at *Perez-Lopez, 2025*). Plots were generated using Chiplot (https://www.chiplot.online/) and ggplot2 (*Wickham, 2016*).

## Sequence alignment

Pairwise alignment of protein sequences was done by EMBOSS Needle (*Rice et al., 2000*). Clustal Omega was used to generate multiple protein sequence alignment (*Sievers and Higgins, 2018*). Kalign 3.4.0 (*Lassmann, 2020*) was used to generate alignment of extremely divergent Mig1 cluster.

## Selection pressure analysis and structure visualization

Coding sequences (CDS) were obtained from the Ensembl Fungi/ENA database. Multiple nucleotide sequences were aligned using the Kc-Align codon-aware aligner (*Nicholas, 2020*). Positions with

more than 50% gaps were removed using the Clipkit (*Steenwyk et al., 2020*) online tool in 'gappy' mode. The trimmed alignments were manually analyzed using Geneious (http://www.geneious.com/) for correct codon alignment. The resulting alignment was uploaded to the Datamonkey server (http://www.datamonkey.org/), which hosts the HyPhy package (*Pond et al., 2005*). All the branches were used as input for FEL (*Kosakovsky Pond and Frost, 2005*) to identify sites under purifying selection (p<0.01). The ESPript 3.0 web server was used to generate multiple sequence alignments (*Robert and Gouet, 2014*). Multiple structures were aligned using mTm-Align (*Dong et al., 2018*). PyMOL 3.0.4 (*Schrödinger, 2015*) was used to visualize the PDB files and color conserved sites or disulfide bridges.

## Expression analysis

Datasets for the RNA-Seq reads obtained at 16- and 26 dpi during *P. brassicae* infection were downloaded from the EBI server (accession number PRJEB12261). The reads from the infected samples were mapped to the *A. thaliana* genome TAIR10 (Genbank accession number GCF_000001735.4) using HiSAT2 (*Kim et al., 2019*) to remove host contaminant sequences. To use Salmon (*Patro et al., 2017*), an index file was created by concatenating the *P. brassicae* genome and CDS sequences. The remaining RNA-Seq reads were quasi-mapped to the index using Salmon 1.10.0 to generate normalized transcripts per million (TPM) counts for all 10,521 genes. TPM values from three replicates were averaged. Pre-processed gene expression data for *U. maydis* was publicly available (*Lanver et al., 2018*).

## Motif scanning

MEME 5.5.5 was used to scan the list of amino acid sequences in '-anr' mode (E<0.1) to discover motifs of any length and frequency. MAST 5.5.5 was used to protein sequences with MEME motifs. Sequence profiles of *P. brassicae* and *S. subterranea* ANK motifs were aligned using Tomtom 5.5.5 (*Gupta et al., 2007*).

## Structural homology search

FoldSeek (https://search.foldseek.com/search) was used to search for structural homology against Uniprot50, Swiss-Prot, the AlphaFold proteome, and PDB. The AFDB cluster database (https://cluster.foldseek.com/) was searched to find cluster members in the AlphaFold database.

## In silico protein-protein interaction prediction

ANK proteins were screened for interaction against a list of *A. thaliana* immune genes using Alpha-Pulldown v1.0.4 (*Yu et al., 2023*). It utilizes AlphaFold Multimer but separates the CPU and GPU jobs and reuses the MSA to reduce compute time. The full AlphaFold 2.3.0 database was used for MSA creation. The resulting models from the AlphaPulldown run were parsed with the supplied singularity image alpha-analysis_jax_0.4.sif to produce the final iPTM +pTM score table. ChimeraX 1.8 (*Meng et al., 2023*) was used to visualize the predicted aligned error for residue pairs under 4 Ångströms at the interface between the two chains produced by AlphaFold-Multimer.

## Yeast two-hybrid assay

Yeast two-hybrid (Y2H) screenings were performed by Hybrigenics Services (https://hybrigenics-services.com/). The coding sequence of *Plasmodiophora brassicae* PBTT_00818 (amino acids 1–490) was synthesized by Twist Biosciences (https://www.twistbioscience.com) and cloned into the pTwist ENTR vector. This construct was then used as a template to amplify and subclone the gene into the pB66 vector as a C-terminal fusion to the Gal4 DNA-binding domain (Gal4-bait). All constructs were verified by sequencing. The bait construct was used to screen a random-primed *Arabidopsis thaliana* seedling (1-week-old) cDNA library [ATH], cloned into the pP6 vector. For the PBTT_00818 screen, approximately 60 million clones, equivalent to six times the complexity of the library, were screened using a mating-based approach with the CG1945 yeast strain, following previously described protocols (*Fromont-Racine et al., 1997*). A total of 98 His[+] colonies were selected on selective medium lacking tryptophan, leucine, and histidine and supplemented with 50 mM 3-aminotriazole (3-AT) to suppress bait autoactivation. Prey fragments from positive colonies were PCR-amplified and sequenced at both

5' and 3' ends, and the resulting sequences were used to identify corresponding proteins via automated BLAST searches against the GenBank (NCBI) database.

For one-by-one (1-by-1) Y2H assays, bait and prey constructs were transformed separately into CG1945 and YHGX13 yeast strains, respectively. Interaction tests were performed using the HIS3 reporter gene system, with growth assays on selective media. As negative controls, bait plasmids were tested with empty prey vectors (pP7), and prey plasmids with empty bait vectors (pB66). The SMAD–SMURF interaction served as a positive control (*Colland et al., 2004*). Each interaction and control was assessed using streaks of three independent yeast clones. Two selective media were used: DO-2 (lacking tryptophan and leucine) served as a control to confirm the presence of both bait and prey plasmids, while DO-3 (lacking tryptophan, leucine, and histidine) was used to detect protein–protein interactions. Increasing concentrations of 3-AT were added to the DO-3 plates to enhance stringency and reduce false positives due to bait autoactivation. All 1-by-1 Y2H experiments were performed in triplicate to ensure reproducibility.

## BiFC assay

Bimolecular fluorescence complementation (BiFC) assays were performed by PronetBio (https://www.pronetbio.com). The coding sequences of *P. brassicae* PBTT_00818 and *Arabidopsis* genes AT3G45640 and AT3G56460 were synthesized and subcloned into the BiFC expression vectors pCAMBIA1300-nYFP and pCAMBIA1300-cYFP, respectively. Recombinant plasmids were first transformed into *Escherichia coli* TOP10 cells for propagation and then introduced into *Agrobacterium tumefaciens* strain GV3101 via electroporation.

*Nicotiana benthamiana* plants (4–6 weeks old, five-leaf stage) were cultivated under greenhouse conditions with a 14 hr light/10 hr dark photoperiod (22–25°C Day / 18–20°C night). For transient expression, healthy leaves (3rd to 6th from the apex) were infiltrated with 30–50 μL of Agrobacterium suspension per site using a needleless syringe to apply the mixture to the abaxial surface. Following infiltration, plants were incubated for 36–48 hr before leaf samples were excised from the marked infiltration zones. Fluorescence signals indicative of protein–protein interaction were detected and imaged using a laser scanning confocal microscope. All BiFC experiments were performed in triplicate to ensure reproducibility. In each assay, the positive control used was the interactive pair HAI1–MPK6 as previously described (*Mine et al., 2017*).

## Acknowledgements

We are grateful to the bioinformatics support personnel and infrastructure at IBIS, Université Laval, for their constant assistance throughout this project, and to Prof. Sylvain Raffaele for providing a dataset used in the study. We also thank Elisa Fantino and Anne-Sophie Brochu for their support in coordinating the interaction validations with PronetBio and Hybrigenics Services. This work was funded by the Canola Agronomic Research Program (Grant ID 2021.4), Western Grain Research Foundation, Canola Council of Canada, Alberta Canola, and Manitoba Canola Growers Association, as well as by the Discovery Program (Grant ID RGPIN-2021–02518) of the Natural Sciences and Engineering Research Council of Canada. We are also thankful to the FRQ – Nature et technologies division – for supporting MAJ and JW through doctoral scholarships.

## Additional information

### Funding

| Funder | Grant reference number | Author |
| --- | --- | --- |
| Canola Council of Canada | 2021.4 | Edel Perez-Lopez |
| Natural Sciences and Engineering Research Council of Canada | RGPIN-2021-02518 | Edel Perez-Lopez |
| Western Grain Research Foundation | | Edel Perez-Lopez |

| Funder | Grant reference number | Author |
|---|---|---|
| Manitoba Canola Growers Association | | Edel Perez-Lopez |
| Alberta Canola Producers Commission | | Edel Perez-Lopez |
| Fonds de recherche du Québec | Doctoral scholarships | Muhammad Asim Javed Jiaxu Wu |

The funders had no role in study design, data collection and interpretation, or the decision to submit the work for publication.

## Author contributions

Soham Mukhopadhyay, Conceptualization, Formal analysis, Investigation, Methodology, Writing – original draft, Writing – review and editing; Muhammad Asim Javed, Investigation, Visualization, Writing – review and editing; Jiaxu Wu, Validation, Investigation, Writing – review and editing; Edel Perez-Lopez, Conceptualization, Supervision, Funding acquisition, Methodology, Writing – original draft, Writing – review and editing

## Author ORCIDs

Soham Mukhopadhyay ⓘ https://orcid.org/0000-0002-5279-0396
Muhammad Asim Javed ⓘ https://orcid.org/0000-0003-1658-3565
Edel Perez-Lopez ⓘ https://orcid.org/0000-0002-3708-8558

Reviewer #1 (Public review): https://doi.org/10.7554/eLife.105185.3.sa1
Reviewer #2 (Public review): https://doi.org/10.7554/eLife.105185.3.sa2
Author response https://doi.org/10.7554/eLife.105185.3.sa3

# Additional files

## Supplementary files

Supplementary file 1. Detailed information about the pathogens investigated in this study and their secretome composition with pLDDT scores, IUPred3 results, and InterproScan identity.

Supplementary file 2. Known/orphan effector families searched in gall-forming secretome investigated in this study.

Supplementary file 3. Structural clusters generated in this study; ID of the top 26 clusters detected; and frequency of clusters, orphan effectors and known fungal effectors per species studied here.

Supplementary file 4. Detailed information on cluster 2, FoldSeek search results, AFDB cluster web tool, and sequence identity matrix with HopQ1.

Supplementary file 5. Sequence identity matrix of *Ustilago maydis* Mig1 protein with other members of cluster 30.

Supplementary file 6. Expression of *Ustilago maydis mig1* and members of the cluster 30 in axenic, 1, 2, 4, 6, 8, and 12 dpi.

Supplementary file 7. Detailed information on the new SUSS effector families identified in this study. Structure-based and Sequence-based clustering of CCG effectors. Structure-based and Sequence-based clustering of AvrSen1-like effectors. MEME output of CCG and RAYH motif scan. Selection of one representative member (highest pLDDT) from each sequence-based cluster for selection pressure analysis in CCGs. Selection of one representative member (highest pLDDT) from each sequence-based cluster to examine RAYH motif conservation in AvrSen1-like cluster.

Supplementary file 8. Sequence clusters generated in this study.

Supplementary file 9. Detailed information of proteins containing repeats. Ankyrin repeat-containing proteins, leucine-rich repeats, and ankyrin repeat-containing proteins with SPK1 domains in *P. brassicae* and *S. subterranean*. Expression of *P. brassicae* ankyrin repeat-containing proteins and leucine-rich repeats at 16- and 26 dpi in *A. thaliana*. *P. brassicae* and *S. subterranean* ankyrin repeat-containing proteins MEME scan.

Supplementary file 10. List of *A. thaliana* and *P. brassicae* proteins used to study modeled

interactions with AlphaFold-Multimer.

Supplementary file 11. Output of the AlphaFold-Multimer predicting interaction between *A. thaliana* immunity hubs and *P. brassicae* Ankyrin repeat-containing proteins.

Supplementary file 12. Detailed list of PBTT_00818 His + clones resulted from a yeast two-hybrid screen versus an *Arabidopsis* seedling library.

MDAR checklist

## Data availability

The datasets used in this study can be downloaded from Zenodo and the scripts from GitHub (copy archived *Perez-Lopez, 2025*).

The following dataset was generated:

| Author(s) | Year | Dataset title | Dataset URL | Database and Identifier |
|---|---|---|---|---|
| Mukhopadhyay S, Javed A, Wu J, Pérez-López E | 2024 | Dataset related to "Structure-guided secretome analysis of gall-forming microbes offers insights into effector diversity and evolution" | https://doi.org/ 10.5281/zenodo. 11152389 | Zenodo, 10.5281/ zenodo.11152389 |

The following previously published datasets were used:

| Author(s) | Year | Dataset title | Dataset URL | Database and Identifier |
|---|---|---|---|---|
| Schwelm A | 2018 | Spongospora subterranea, whole genome shotgun sequencing project | https://www.ncbi. nlm.nih.gov/ nuccore/ OUQQ00000000 | NCBI Nucleotide, OUQQ00000000 |

*Continued on next page*

*Continued*

| Author(s) | Year | Dataset title | Dataset URL | Database and Identifier |
|---|---|---|---|---|
| Kamper J, Kahmann R, Bolker M, Brefort T, Saville BJ, Banuett F, Kronstad JW, Gold SE, Muller O, Perlin MH, Wosten HA, de Vries R, Ruiz-Herrera J, Reynaga-Pena CG, Snetselaar K, McCann M, Perez-Martin J, Feldbrugge M, Basse CW, Steinberg G, Ibeas JI, Holloman W, Guzman P, Farman M, Stajich JE, Sentandreu R, Gonzalez-Prieto JM, Kennell JC, Molina L, Schirawski J, Mendoza-Mendoza A, Greilinger D, Munch K, Rossel N, Scherer M, Vranes M, Ladendorf O, Vincon V, Fuchs U, Sandrock B, Meng S, Cahill MJ, Boyce KJ, Klose J, Klosterman SJ, Deelstra HJ, Ortiz-Castellanos L, Li W, Sanchez-Alonso P, Schreier PH, Hauser-Hahn I, Vaupel M, Koopmann E, Friedrich G, Voss H, Schluter T, Margolis J, Platt D, Swimmer C, Gnirke A, Chen F, Vysotskaia V, Mannhaupt G, Guldener U, Munsterkotter M, Haase D, Oesterheld M, Mewes HW, Mauceli EW, DeCaprio D, Wade CM, Butler J, Young S, Jaffe DB, Calvo S, Nusbaum C, Galagan J, Birren BW, Ma LJ, Ho EC | 2006 | Mycosarcoma maydis strain 521, whole genome shotgun sequencing project | https://www.ncbi.nlm.nih.gov/nuccore/AACP00000000 | NCBI Nucleotide, AACP00000000 |

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
