## [Editor Report · eLife Assessment]

This study presents an **important** discovery regarding the diversity and evolution of gall-forming microbial effectors. Supported by **convincing** computational structural predictions and analyses, the research provides insights into the unique mechanisms by which gall-forming microbes exert their pathogenicity in plants. This study also offers guidance that is of value for future studies on pathogen effector function and co-evolution with host plants.

---

## [Referee Report · Reviewer #1 (Public review)]

Summary:

This manuscript presents a comprehensive structure-guided secretome analysis of gall-forming microbes, providing valuable insights into effector diversity and evolution. The authors have employed AlphaFold2 to predict the 3D structures of the secretome from selected pathogens and conducted a thorough comparative analysis to elucidate commonalities and unique features of effectors among these phytopathogens.

Strengths:

The discovery of conserved motifs such as 'CCG' and 'RAYH' and their central role in maintaining the overall fold is an insightful finding. Additionally, the discovery of a nucleoside hydrolase-like fold conserved among various gall-forming microbes is interesting.

Weaknesses:

Important conclusions are not verified by experiments.

Comments on revisions: I acknowledge the authors' revision efforts.

---

## [Referee Report · Reviewer #2 (Public review)]

Summary:

Soham Mukhopadhyay et al. investigated the protein folding of the secretome from gall-forming microbes using the AI-based structure-modeling tool AlphaFold2. Their study analyzed six gall-forming species, including two Plasmodiophorid species and four others spanning different kingdoms, along with one non-gall-forming Plasmodiophorid species, Polymyxa betae. The authors found no effector fold specifically conserved among gall-forming pathogens, leading to the conclusion that their virulence strategies are likely achieved through diverse mechanisms. However, they identified an expansion of the Ankyrin repeat family in two gall-forming Plasmodiophorid species, with a less pronounced presence in the non-gall-forming Polymyxa betae. Additionally, the study revealed that known effectors such as CCG and AvrSen1 belong to sequence-unrelated but structurally similar (SUSS) effector clusters.

Strengths:

(1) The bioinformatics analyses presented in this study are robust, and the AlphaFold2-derived resources deposited in Zenodo provide valuable resources for researchers studying plant-microbe interactions. The manuscript is also logically organized and easy to follow.

(2) The inclusion of the non-gall-forming Polymyxa betae strengthens the conclusion that no effector fold is specifically conserved in gall-forming pathogens and highlights the specific expansion of the Ankyrin repeat family in gall-forming Plasmodiophorids.

(3) Figure 4a and 4b effectively illustrate the SUSS effector clusters, providing a clear visual representation of this finding.

(4) Figure 1 is a well-designed, comprehensive summary of the number and functional annotations of putative secretomes in gall-forming pathogens. Notably, it reveals that more than half of the analyzed effectors lack known protein domains in some pathogens, yet some were annotated based on their predicted structures, despite the absence of domain annotations.

Weaknesses:

(1) The effector families discussed in this paper remain hypothetical in terms of their functional roles, which is understandable given the challenges of demonstrating their functions experimentally. However, this highlights the need for experimental validation as a next step.

Authors' response: Thank you. Yes, there is a lot of work to do in the coming years.

Reviewer's response: Incorporating experimental validation substantially strengthened the manuscript. Did you try the AlphaFold-Multimer prediction of the interaction between PBTT_00818 and the GroES-like protein? Does the model indicate a high-confidence interface?

(2) Some analyses, such as those in Figure 4e, emphasize motifs derived from sequence alignments of SUSS effector clusters. Since these effectors are sequence-unrelated, sequence alignments might be unreliable. It would be more rigorous to perform structure-based alignments in addition to sequence-based ones for motif confirmation. For instance, methods described in Figure 3E of de Guillen et al. (2015, https://doi.org/10.1371/journal.ppat.1005228) or tools like Foldseek could be useful for aligning structures of multiple sequences.

Authors' response: In Fig. 4e, we highlight the conserved cysteine residues. While there is no clearly conserved overall motif, the figure illustrates that despite the high sequence divergence, the key cysteines involved in disulfide-bridge formation are consistently conserved across the sequences.

Reviewer's response: Understood. Nevertheless, if a reliable sequence alignment can indeed be generated, I would interpret this to mean that the CCG effectors constitute a highly diversified family rather than being truly sequence unrelated. By comparison, members of the MAX effector family share a common fold, yet their sequences are so divergent that sequence alignment is impossible.

(3) When presenting AlphaFold-generated structures, it is essential to include confidence scores such as pLDDT and PAE. For example, in Figure 1D of Derbyshire and Raffaele (2023, https://doi.org/10.1038/s41467-023-40949-9), the structural representations were colored red due to their high pLDDT scores, emphasizing their reliability.

Authors' response: Thank you for the observation. Due to the restrictive parameters used in our analysis, over 90 % of the structure would appear red. For this reason, we chose not to include the color scale, as it would not provide additional informative value in this context.

Reviewer's response: Understood.

---

## [Author Response]

The following is the authors’ response to the original reviews.

**Reviewer #1 (Public review):**
Summary:This manuscript presents a comprehensive structure-guided secretome analysis of gall-forming microbes, providing valuable insights into effector diversity and evolution. The authors have employed AlphaFold2 to predict the 3D structures of the secretome from selected pathogens and conducted a thorough comparative analysis to elucidate commonalities and unique features of effectors among these phytopathogens.Strengths:The discovery of conserved motifs such as 'CCG' and 'RAYH' and their central role in maintaining the overall fold is an insightful finding. Additionally, the discovery of a nucleoside hydrolase-like fold conserved among various gall-forming microbes is interesting.Weaknesses:Important conclusions are not verified by experiments.

Thank you very much. There are many aspects of this study that could be further validated, each potentially requiring years of work. Therefore, we chose to focus on two specific hypotheses: are AlphaFol-Multimer predictions accurate? Can ANK target more than one host protein? Particularly, we focused on the identification of putative targets for one of the ankyrin repeat proteins, PBTT_00818 (Fig. 6). Using one-by-one yeast two-hybrid (Y2H) assays, we tested the AlphaFold-Multimer prediction of an interaction between PBTT_00818 and MPK3. The interaction did not occur in yeast, suggesting it might not take place under those conditions.

This negative result led us to perform a Y2H screen using an Arabidopsis cDNA library, which identified a GroES-like protein, highly expressed in roots, as a potential target of the ANK effector. Surprisingly, both the PBTT_00818–MPK3 and PBTT_00818–GroES-like protein interactions were later confirmed in planta using BiFC assays. These findings suggest two key points: (1) AlphaFold predictions can be accurate for ANK proteins, and (2) ANK domains, known for mediating protein-protein interactions, may enable these effectors to target multiple host proteins.

Although the precise biological implications remain unclear, it is possible that ANK proteins act as scaffolds or adaptors for other effectors during infection. The validations presented here open exciting avenues for further research into the role of ANK proteins in Plasmodiophorid pathogenesis and gall formation. This is presented in the corrected preprint and Fig. 7, Table S12, Fig. S7-S8.

**Reviewer #2 (Public review):**
Summary:Soham Mukhopadhyay et al. investigated the protein folding of the secretome from gall-forming microbes using the AI-based structure modeling tool AlphaFold2. Their study analyzed six gall-forming species, including two Plasmodiophorid species and four others spanning different kingdoms, along with one non-gall-forming Plasmodiophorid species, Polymyxa betae. The authors found no effector fold specifically conserved among gall-forming pathogens, leading to the conclusion that their virulence strategies are likely achieved through diverse mechanisms. However, they identified an expansion of the Ankyrin repeat family in two gall-forming Plasmodiophorid species, with a less pronounced presence in the non-gall-forming Polymyxa betae. Additionally, the study revealed that known effectors such as CCG and AvrSen1 belong to sequence-unrelated but structurally similar (SUSS) effector clusters.Strengths:(1) The bioinformatics analyses presented in this study are robust, and the AlphaFold2-derived resources deposited in Zenodo provide valuable resources for researchers studying plant-microbe interactions. The manuscript is also logically organized and easy to follow.(2) The inclusion of the non-gall-forming Polymyxa betae strengthens the conclusion that no effector fold is specifically conserved in gall-forming pathogens and highlights the specific expansion of the Ankyrin repeat family in gall-forming Plasmodiophorids.(3) Figure 4a and 4b effectively illustrate the SUSS effector clusters, providing a clear visual representation of this finding.(4) Figure 1 is a well-designed, comprehensive summary of the number and functional annotations of putative secretomes in gall-forming pathogens. Notably, it reveals that more than half of the analyzed effectors lack known protein domains in some pathogens, yet some were annotated based on their predicted structures, despite the absence of domain annotations.Weaknesses:(1) The effector families discussed in this paper remain hypothetical in terms of their functional roles, which is understandable given the challenges of demonstrating their functions experimentally. However, this highlights the need for experimental validation as a next step.

Thank you. Yes, there is a lot of work to do in the coming years.

(2) Some analyses, such as those in Figure 4e, emphasize motifs derived from sequence alignments of SUSS effector clusters. Since these effectors are sequence-unrelated, sequence alignments might be unreliable. It would be more rigorous to perform structure-based alignments in addition to sequence-based ones for motif confirmation. For instance, methods described in Figure 3E of de Guillen et al. (2015, https://doi.org/10.1371/journal.ppat.1005228) or tools like Foldseek could be useful for aligning structures of multiple sequences.

In Fig. 4e, we highlight the conserved cysteine residues. While there is no clearly conserved overall motif, the figure illustrates that despite the high sequence divergence, the key cysteines involved in disulfide bridge formation are consistently conserved across the sequences.

(3) When presenting AlphaFold-generated structures, it is essential to include confidence scores such as pLDDT and PAE. For example, in Figure 1D of Derbyshire and Raffaele (2023, https://doi.org/10.1038/s41467-023-40949-9), the structural representations were colored red due to their high pLDDT scores, emphasizing their reliability.

Thank you for the observation. Due to the restrictive parameters used in our analysis, over 90% of the structure would appear red. For this reason, we chose not to include the color scale, as it would not provide additional informative value in this context.

**Reviewer #1 (Recommendations for the authors):**
Experimental validation of the significance of 'CCG' and 'RAYH' motifs would further strengthen this study.Regarding the Mig1-like protein in *Ustilago maydis*, the presence of four conserved cysteine residues that are pivotal for maintaining the stability of its folded structure raises an intriguing question. Specifically, while many Mig cluster effectors contain four cysteine residues that form two conserved disulfide bridges, this structure is notably absent in the Mig protein itself. The author has speculated that these four cysteine residues form two conserved disulfide bonds, which are crucial for the stability of Mig protein folding. However, this hypothesis remains unvalidated. To test this prediction, it would be prudent to simulate mutations in the cysteine residues corresponding to the disulfide bonds in Mig and employ molecular dynamics simulations to assess the stability of folding before and after the mutation.

Mig-1 does contain the four conserved cysteine residues responsible for forming disulfide bridges. However, due to the high divergence among Mig-1-like sequences, the alignment software was unable to properly align all the cysteine residues. As a result, Mig-1 may appear to lack these conserved cysteines in the alignment, although they are indeed present upon individual inspection. This is an area that research groups working with U. maidis as a model could explore further to expand our understanding of this effector family.

Could you please clarify why talking about Ankyrins and LRR in *Arabidopsis thaliana* (line 252)? Additionally, what are the structural and functional differences between the LRR sequences of P. brassicae and those of the host plants?

This sentence refers to the identification of the ANK motif in P. brassicae and S. spongospora, not in *Arabidopsis thaliana*. While the hydrophobic core of the ANK domains appears conserved between the host and the pathogen, the surface residues are highly polymorphic.

The evidence supporting the interaction between the ANK effector and Arabidopsis immunity-related proteins, as validated using AlphaFold-Multimer, is currently limited. To enhance the reliability of these data, it is advisable for the author to select several pairs of proteins predicted to interact for further experimental verification.

We conducted a large-scale yeast two-hybrid (Y2H) screen using the ANK domain effector PBTT_00818, which was selected due to its high iPTM+pTM score. The Y2H interactions were subsequently validated through BiFC assays. Our results show that PBTT_00818 interacts with Arabidopsis MPK3 in the nucleus, consistent with predictions from the AlphaFold2-multimer model. In addition, PBTT_00818 was also found to target AT3G56460, a GroES-like zinc-binding alcohol dehydrogenase, also localized in the nucleus.

While the manuscript is well-composed, certain sections could be enhanced for clarity and readability. For example, the discussion section could be expanded to include a more in-depth analysis of the implications of the findings for understanding the virulence mechanisms of gall-forming microbes. Additionally, a comparison of the findings with previous studies on related pathogens would provide a more comprehensive perspective.

Certain sections of the discussion have been expanded. However, we chose to focus on the novel aspects of the study and to avoid comparisons with other plant pathogens, as those mechanisms are already well known and extensively studied. Studies using AlphaFold in plant pathology are also limited.

****Reviewer #2 (Recommendations for the authors):****The results of clustering analyses are highly dependent on the chosen thresholds. Given that the authors provide clear and well-designed visualizations of SUSS effectors in Figures 4a and 4b, applying the same presentation methods to Figures 5a and 5b could make these analyses more convincing.

We were able to generate the all-vs-all matrix for Figures 4a and 4b because it involved only 13 proteins. However, Figure 5b includes over 40 effectors, making it impractical to visualize the data in the same way. Instead, we presented the sequence-based clusters as nodes and connected them based on structural similarity.